# Noradrenaline blockade specifically enhances metacognitive performance

Tobias U Hauser[1,2*†], Micah Allen[1,3†], Nina Purg[1], Michael Moutoussis[1,2], Geraint Rees[1,3], Raymond J Dolan[1,2]

[1]Wellcome Trust Centre for Neuroimaging, University College London, London, United Kingdom; [2]Max Planck University College London Centre for Computational Psychiatry and Ageing Research, London, United Kingdom; [3]Institute of Cognitive Neuroscience, University College London, London, United Kingdom

**Abstract** Impairments in metacognition, the ability to accurately report one's performance, are common in patients with psychiatric disorders, where a putative neuromodulatory dysregulation provides the rationale for pharmacological interventions. Previously, we have shown how unexpected arousal modulates metacognition (Allen et al., 2016). Here, we report a double-blind, placebo-controlled, study that examined specific effects of noradrenaline and dopamine on both metacognition and perceptual decision making. Signal theoretic analysis of a global motion discrimination task with adaptive performance staircasing revealed that noradrenergic blockade (40 mg propranolol) significantly increased metacognitive performance (type-II area under the curve, AUROC2), but had no impact on perceptual decision making performance. Blockade of dopamine D2/3 receptors (400 mg amisulpride) had no effect on either metacognition or perceptual decision making. Our study is the first to show a pharmacological enhancement of metacognitive performance, in the absence of any effect on perceptual decision making. This enhancement points to a regulatory role for noradrenergic neurotransmission in perceptual metacognition.

*For correspondence: t.hauser@ucl.ac.uk

†These authors contributed equally to this work

Competing interests: The authors declare that no competing interests exist.

Making a decision is often accompanied by a conscious feeling of confidence (*Flavell, 1979*). Subjective confidence reports typically show a good correspondence to actual task performance, reflecting a metacognitive ability for accurate introspection (*Fleming et al., 2010*). Impairments in metacognition can compromise decision making and lead to misjudgements of actual performance, as found in several psychiatric dimensions, such as schizophrenia, attention-deficit/hyperactivity disorder or compulsivity (*Frith, 1992*; *Knouse et al., 2005*; *Lysaker et al., 2010*; *Hauser et al., 2017*).

The neurocognitive mechanisms from which confidence, and metacognitive ability in general, arise are ill understood. While classic accounts see confidence as a mere extension of a perceptual sampling process (*Kiani and Shadlen, 2009*; *Pleskac and Busemeyer, 2010*; *Meyniel et al., 2015*; *Moran et al., 2015*), other evidence points to a non-trivial relationship between decision making and confidence that invoke distinct decision making and metacognitive processes (*Fleming et al., 2010*; *Fleming and Dolan, 2012*; *Allen et al., 2016*; *Allen et al., 2017*). Metacognition can thus be understood as incorporating both decision-related and domain-general information (*Fleming and Daw, 2017*) and frontopolar and hippocampal brain structures, for example, have been shown to contribute specifically to metacognition but not to perceptual decision making (*Fleming et al., 2010*, *2012*; *Allen et al., 2017*). In our previous study, we provided evidence that arousal can bias metacognition independently of decision accuracy (*Allen et al., 2016*), in accordance with other studies showing confidence-accuracy dissociations (*Fleming et al., 2015*; *Spence et al., 2016*) and suggested that these biases might be under neuromodulatory control via neural gain (*Eldar et al., 2013*; *Hauser et al., 2016*).

Two candidate neuromodulators likely to affect metacognition are the catecholamines noradrenaline and dopamine. These neurotransmitters have their origin in brainstem nuclei that project broadly to cortical and subcortical regions, including prefrontal cortex and hippocampus (*Hauser et al., 2016*). Both dopamine and noradrenaline contribute to the regulation of arousal and higher-order cognition (*Usher et al., 1999*; *Aston-Jones and Cohen, 2005*; *Yu and Dayan, 2005*; *Pessiglione et al., 2006*; *De Martino et al., 2008*; *Chowdhury et al., 2013*; *Eldar et al., 2013*; *Rigoli et al., 2016*), and their dysregulation is widely inferred to contribute to various manifestations of psychiatric illness (*Yamamoto and Hornykiewicz, 2004*; *Laruelle, 2013*; *Hauser et al., 2016*). However, whether and which of these neuromodulators influences metacognition is unknown.

Here we assessed whether noradrenaline or dopamine have a distinct influence on metacognition, independent of any effect on perceptual decision making. The latter consideration is important because differences in perceptual decision making can distort an assessment of metacognition (*Fleming and Lau, 2014*). Consequently, we used an approach where we kept perceptual decision making equivalent across subjects, using a staircase procedure. This, combined with a signal detection analysis (*Green and Swets, 1966*; *Galvin et al., 2003*), enabled us to measure drug effects on metacognition while controlling for potential effects on perceptual decision making. This overcomes a limitation of a previous study that assessed the impact of dopamine on confidence (*Lou et al., 2011*).

To examine the influence of noradrenaline and dopamine we employed two pharmacological manipulations. Many pharmacological agents have high affinity for both dopaminergic and noradrenergic receptors and synaptic function. On this basis, we selected drugs with selective high affinity, the $\beta$-adrenoceptor antagonist propranolol in the case of noradrenaline, and the D2/3 receptor antagonist amisulpride in the case of dopamine. In a double-blind, placebo-controlled design we demonstrate that the noradrenergic agent propranolol uniquely improves metacognition in the absence of an effect on perceptual performance, with no effect seen following administration of the dopamine antagonist amisulpride.

## Results

### Noradrenaline blockade modulates metacognition

To examine effects of noradrenaline and dopamine (versus placebo) on metacognition we performed a double-blind, between-subjects, placebo-controlled study. Each of the three groups consisted of 20 subjects matched for gender, age, affect (*Watson et al., 1988*), and intellectual abilities (*Wechsler, 1999*) (*Table 1*). Due to differences in their pharmacokinetic properties we administered active drugs orally at two different time points. The dopamine group received 400 mg of amisulpride (selective D2/3 antagonist) 110 min prior to a metacognition task and an additional placebo 30 min

**Table 1.** Group characteristics. The three groups did not differ in their gender, age, intellectual abilities (IQ) (*Wechsler, 1999*) and their positive and negative affective states before and after drug administration. PANAS: positive and negative affective schedule (*Watson et al., 1988*), PA: positive affect, NA: negative affect, pre: before drug administration, post: after drug administration (mean±SD).

|  | Placebo | Propranolol | Amisulpride |  |
|---|---|---|---|---|
| gender (f/m) | 10/10 | 10/10 | 10/10 |  |
| age | 24.50 ± 4.16 | 23.15 ± 4.31 | 22.35 ± 2.21 | $F_{(2,57)}=1.74$, p=0.185 |
| IQ | 112.45 ± 12.22 | 118.75 ± 8.55 | 114.60 ± 11.77 | $F_{(2,57)}=1.70$, p=0.191 |
| PANAS PA pre | 31.15 ± 10.08 | 27.70 ± 8.28 | 28.90 ± 6.60 | $F_{(2,57)}=0.86$, p=0.428 |
| PANAS NA pre | 11.70 ± 2.23 | 13.55 ± 5.48 | 13.10 ± 3.23 | $F_{(2,57)}=1.23$, p=0.300 |
| PANAS PA post | 29.22 ± 10.47 | 27.15 ± 7.75 | 27.80 ± 8.12 | $F_{(2,57)}=0.286$, p=0.752 |
| PANAS NA post | 11.45 ± 2.37 | 11.95 ± 4.87 | 11.25 ± 1.92 | $F_{(2,57)}=0.236$, p=0.790 |

after the amisulpride administration (*Figure 1A*). The noradrenaline group received a placebo at 110 min prior to the task and then 40 mg of propranolol (non-selective $\beta$-adrenoceptor antagonist) 30 min after placebo administration. The placebo group received placebo at the both time points to match the administration schedules of the other groups. A post-experiment evaluation revealed that subjects were not aware of whether and which drug they received ($\chi^2(4)=1.26$, p=0.868; missing data from two subjects). There were no effects of drug on mood (PANAS; *Watson et al., 1988*) ratings (main effect of drug: $F(2,57)=.16$, p=0.852; time x drug: $F(2,57)=.19$, p=0.827; time x affect x drug: $F(2,57)=2.17$, p=0.124).

Eighty minutes after the second drug administration, subjects performed a visual global motion discrimination task that included confidence judgements (*Figure 1B*). Subjects decided whether the overall motion of a short burst of randomly moving dots was directed to the left or right of the vertex. Subsequently, and in the absence of feedback on whether they were correct or not, they indicated confidence in their decision on that trial using a sliding visual analogue scale. To control for potential drug effects on perceptual performance, we matched the subjects' decision accuracy by continuously adapting the global motion orientation using a staircase procedure (*Cornsweet, 1962*).

To examine metacognitive abilities, we analysed type-II performance as derived from signal detection theory (*Green and Swets, 1966*; *Galvin et al., 2003*). This measures subjects' awareness into their own performance by assessing how well their confidence ratings match their true accuracy (i.e., 'how much more confident am I if I make a correct vs incorrect decision'). We calculated type-II area under the receiver-operating-characteristics curve (AUROC2) (*Fleming et al., 2010*, *2012*; *Weil et al., 2013*; *Allen et al., 2017*) for each subject and then compared this metric between groups using an ANOVA with drug group as a between-subjects factor. The analysis revealed a significant effect of drug on AUROC2 (*Figure 2*, $F(2,55)=5.192$, p=0.009, $\eta^2=0.16$), with follow up

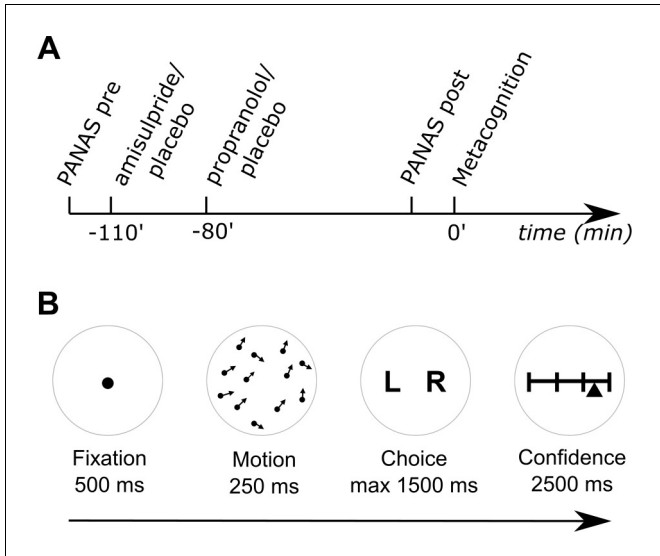

**Figure 1.** Experimental design and metacognition task. (**A**) After filling out a baseline mood questionnaire (PANAS pre), subjects received two different drugs 110 and 80 min prior to the metacognition task. A dopamine subject group first received 400 mg amisulpride (dopamine D2/3 receptor antagonist) and subsequently placebo, whereas the noradrenaline group first received placebo and then 40 mg propranolol ($\beta$-adrenoceptor antagonist). Subjects of a placebo group received placebo at both times. Eighty minutes after the second drug administration, subjects filled out a second mood questionnaire (PANAS post) and then performed a metacognition task. (**B**) To assess subjects' metacognitive abilities, we used a global motion discrimination task with subsequent confidence judgements. After a fixation period, subjects saw 1100 dots moving randomly with an average motion pointing either to the left or right. After 250 ms, subjects had to indicate the overall direction of the moving dots by using keyboard arrows. Subsequently, they indicated their confidence about their decision using a sliding visual analogue scale. Subjects were instructed to use the full width of the scale by indicating high confidence on the right and low confidence on the left side of the scale.

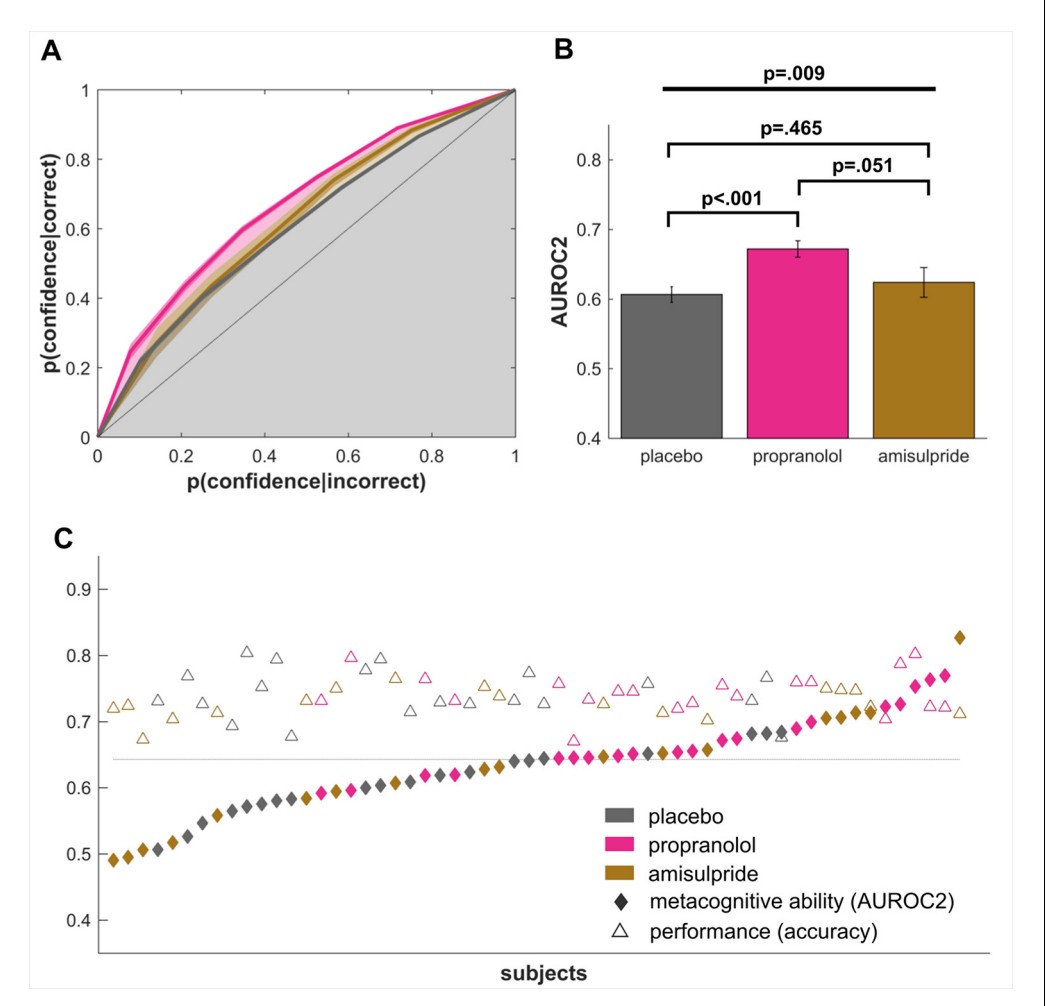

**Figure 2.** Propranolol improves metacognitive abilities. (**A**) Signal detection theoretic analysis revealed a significantly increased metacognitive ability, as measured by the type-II area under the ROC curve (AUROC2). (**B**) A highly significant effect of propranolol compared to placebo shows that propranolol increases metacognitive abilities. The difference between propranolol and amisulpride suggests that this performance increase might be specific to an influence on noradrenaline but not dopamine function. (**C**) Individual AUROC2 metrics show that most subjects in the propranolol group perform above the median metacognitive performance (dotted line), while perceptual decision making performance was relatively stable across all groups. mean ±1 SEM; fat line: ANOVA; square brackets: t-tests.

The following source data is available for figure 2:

**Source data 1.** Source data for *Figures 2* and *3*.

t-tests showing the propranolol group performed significantly better than a placebo group (t(38) =4.00, p<0.001, *d* = 1.26). The propranolol group also performed marginally better than an ami-sulpride group (t(36)=2.02, p=0.051, *d* = 0.65) with the latter having an equal performance as the placebo group (t(36)=.74, p=0.465, *d* = 0.23). To evaluate evidence for this null effect in the ami-sulpride group, we additionally performed a Bayesian two-sample t-test comparing the placebo and amisulpride groups (*Rouder et al., 2009*; *Dienes, 2014*). This analysis revealed a Bayes Factor of 3.31, corresponding to moderate evidence for the null hypothesis. These results indicate that inhibi-tion of noradrenergic function improves metacognitive insight, in the absence of any effect of amisulpride.

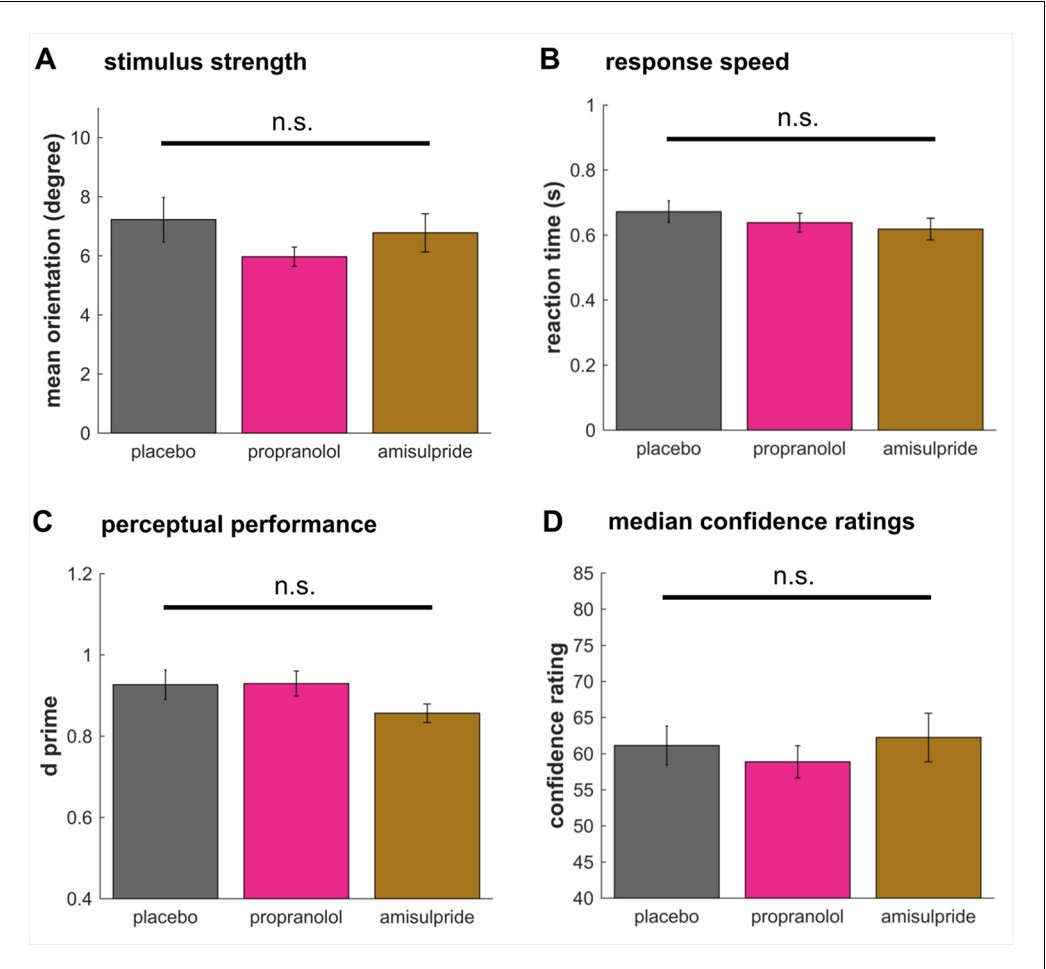

**Figure 3.** Drug effects on perceptual decision making. No drug effects were observed on the signal strength (**A**, stimulus motion orientation) or the response speed (**B**). In line with no difference in accuracy, perceptual sensitivity *d'* did not differ between groups (**C**). Median confidence ratings (**D**) showed no difference revealing that there was no bias in the average rating behaviour between groups. These findings suggest that noradrenaline blockade selectively boosts metacognitive sensitivity in the absence of any effect on perceptual decision making. mean ±1 SEM; n.s. p>0.10.

## Improved metacognition mainly driven by confidence on error trials

To further understand the noradrenaline-induced metacognitive enhancement, we compared median confidence ratings for correct and incorrect trials between the propranolol and placebo group. A significant group-by-correctness interaction ($F(1,38)=8.66$, $p=0.006$, $\eta^2=0.19$) in the absence of a group main effect ($F(1,38)=1.83$, $p=0.185$, $\eta^2=0.05$) suggests that a confidence rating difference between correct and incorrect trials underlies the observed group differences. Subsequent t-tests demonstrated that this effect was primarily driven by error trials (error trials: $t(38)=2.17$, $p=0.036$, $d = 0.69$, correct trials: $t(38)=-.193$, $p=0.848$, $d = 0.06$), suggesting that the propranolol group exhibited lower confidence for error trials.

## Metacognitive differences are not explained by perceptual performance

Metacognitive measures, as used here, can be influenced by differences in perceptual performance (*Fleming and Lau, 2014*). We deliberately used a staircase procedure to keep performance equivalent between groups (mean accuracy: $F(2,55)=1.60$, $p=0.212$, $\eta^2=0.05$, cf. *Figure 2C*). A signal-detection theoretic analysis confirmed these findings, revealing the absence of any significant

differences in either perceptual sensitivity *d'* (*Figure 3C*, F(2,55)=1.69, p=0.194, $\eta^2$=0.07) or response bias *c* (F(2,55)=2.29, p=0.112, $\eta^2$=0.08) (*Green and Swets, 1966*). To additionally ensure that differences in AUROC2 were not influenced by any of these measures, we also compared AUROC2 using an ANCOVA with *d'*, response bias *c*, and stimulus signal strength (mean orientation) as covariates, revealing the same group difference for AUROC2 after controlling for these potential biases (F(2,51)=4.99, p=0.010, $\eta^2$=0.17).

## No drug effect on perceptual decision making

To test whether perceptual decision making was affected by our drug interventions, we analysed whether stimulus strength, measured by mean stimulus motion orientation, differed between groups. There was no significant difference in stimulus strength (*Figure 3A*, F(2,55)=1.16, p=0.321, $\eta^2$=0.04), indicating perceptual performance was not significantly affected by the drug manipulations. Likewise, there was no drug effect on reaction times (*Figure 3B*, F(2,55)=.87, p=0.424, $\eta^2$=0.03). Lastly, to test whether there were baseline differences in how the groups were utilising the confidence scale, we examined the median confidence ratings, but found no difference (*Figure 3D*, F(2,55)=.38, p=0.684, $\eta^2$=0.01), supporting the result that an enhanced metacognitive ability under propranolol is not due to a bias in use of the confidence rating scale.

## Discussion

Confidence determines how much we trust our decisions and how strongly they influence future behaviour. A read out of confidence in a decision that fails to reflect actual performance will lead to poor decisions and long-term adverse outcomes. Impaired metacognition is reported in psychiatric disorders (*Frith, 1992*; *Knouse et al., 2005*; *Lysaker et al., 2010*; *Wells, 2011*; *Hauser et al., 2017*), and its pharmacological remediation could provide a target for treatment (*Wells, 2011*). Here we show that inhibition of central noradrenaline (by means of propranolol) function enhances perceptual metacognitive ability. A dopamine blockade (by means of amisulpride) had no impact on metacognition and neither drug manipulation had an impact on core perceptual performance.

Noradrenaline is known to impact arousal and higher-order cognition, but the precise mechanisms remain obscure. Influential accounts propose noradrenergic modulation of information processing, either through neural gain (*Aston-Jones and Cohen, 2005*; *Eldar et al., 2013*) or by signalling unexpected uncertainty (*Yu and Dayan, 2005*; *Dayan et al., 2006*). Our finding that blocking noradrenaline leads to improved metacognitive performance can be understood within both frameworks. Metacognition can be thought of as a higher-order process that follows a perceptual decision making stage and integrates perceptual and other sources of information, such as interoceptive states and general arousal (*Allen et al., 2016*), to form an overall confidence judgement. The neural gain hypothesis (*Aston-Jones and Cohen, 2005*; *Eldar et al., 2013*) proposes that noradrenaline amplifies strong and diminishes weak signals throughout the brain, with the effect of an increased 'contrast' between strong and weak signals. Due to a nonlinearity in this amplification it is likely to neglect subtle signal differences, and thus omit the breath and detail of information conveyed. This in turn means noradrenaline might render detailed stimulus properties unavailable to the metacognitive process, impairing the precision of a metacognitive judgement. The latter theory (*Dayan et al., 2006*) suggests phasic noradrenaline is elicited by unexpected uncertainty or arousal, such as when making an erroneous choice (*Ullsperger et al., 2010*). This phasic burst acts by interrupting ongoing processes and leads to a resetting and erasure of currently maintained information to enable an orienting response (*Sokolov et al., 2002*; *Dayan et al., 2006*). In the context of our paradigm, this suggests that following an incorrect response accumulated sensory information is reset and unavailable for confidence judgement, leading to poorer metacognitive performance. This is supported by our finding of a primary drug-effect on confidence in erroneous trials. On both accounts, a blockade of noradrenaline by propranolol hinders the noradrenaline-related loss of information to provide complete perceptual information to a confidence-related process. This is also in line with our previous findings showing that unexpected arousal biases metacognition (*Allen et al., 2016*), an effect possibly modulated by noradrenaline.

In our experiment, perceptual metacognition was solely influenced by manipulation of noradrenaline, and not by blocking dopamine D2/3 receptors. This is of interest as both neuromodulators are often ascribed similar functions including a role in exploration (*Hauser et al., 2016*), neural gain

(*Eldar et al., 2013*; *Fiore et al., 2016*; *Hauser et al., 2016*), salience (*Bromberg-Martin et al., 2010*; *Kahnt and Tobler, 2013*), (prediction) error signalling (*Holroyd and Coles, 2002*; *Hauser et al., 2014*) and effort processing (*Bouret et al., 2012*). Likewise, neural recordings from dopaminergic and noradrenergic brainstem nuclei have revealed surprisingly similar neural response patterns (*Bouret et al., 2012*; *Varazzani et al., 2015*). Our finding of enhanced perceptual metacognition with noradrenaline blockade might reflect a rare sensitivity to the actions of one of these neuromodulators. Our findings also raise the possibility that a recent report of increased performance and confidence ratings following dopaminergic enhancement (*Lou et al., 2011*) may reflect a performance, but not a metacognitive, effect. This finding is akin to a previous report of specific testosterone effects, where there was an effect on perceptual performance but not metacognition (*Wright et al., 2012*).

An important caveat for comparing the amisulpride and propranolol groups directly is that little is known about the precise pharmacokinetics and how comparable the dosage effects are. We took great care in the design of the study to render the two drug conditions as comparable as possible. First, because of the slightly different absorption rates, we administered amisulpride 30 min before propranolol, in keeping with previous drug schedules (*Peretti et al., 1997*; *Ramaekers et al., 1999*; *Strange et al., 2003*; *Silver et al., 2004*; *Strange and Dolan, 2004*; *Hurlemann et al., 2005*; *Alexander et al., 2007*; *Gibbs et al., 2007*; *De Martino et al., 2008*; *Kahnt et al., 2015*; *Kahnt and Tobler, 2017*). Second, to render the cognitive effects of the drugs as similar as possible, we selected dosages that were commonly reported in previous studies of neurocognition (i.e. 40 mg propranolol, 400 mg amisulpride) (e.g., *Ramaekers et al., 1999*; *Strange et al., 2003*; *Silver et al., 2004*; *Strange and Dolan, 2004*; *Hurlemann et al., 2005*; *Alexander et al., 2007*; *Gibbs et al., 2007*; *De Martino et al., 2008*; *Kahnt et al., 2015*; *Kahnt and Tobler, 2017*). However, we know little about the magnitude of these drug effects on the brain. A previous study of sulpiride, which has a similar chemical formulation to amisulpride, but slightly different pharmacokinetics, suggests that a single-dose of 400 mg leads to a occupancy of ~28% of D2 receptors (*Mehta et al., 2008*). Unfortunately, there are no PET studies reporting on single-dose amisulpride, and there are no occupancy studies of propranolol, thus rendering it difficult to directly quantify and compare our dosage effects.

In this study, we show that noradrenaline specifically influences perceptual metacognition but not perceptual decision making. It is interesting to speculate whether our findings are generalizable to metacognition in non-perceptual domains. Recent studies show that a metacognitive ability is relatively stable across different perceptual decision making tasks (*Song et al., 2011*; *McCurdy et al., 2013*), even when probing different sensory modalities (*de Gardelle et al., 2016*; *Garfinkel et al., 2016*). However, it is unclear whether metacognition within different cognitive domains (e.g., perception vs memory) rely on the same processes, with evidence from neuroimaging suggesting that these functions utilise unique neural networks (*Baird et al., 2013*; *Fleming et al., 2014*). Given that noradrenaline modulates activity on a whole-brain level (*Hauser et al., 2016*), it is possible that noradrenergic metacognition effects can be observed in domains other than perception, an interesting area for future studies. Lastly, our previous findings of an embodied reflection of confidence by means of cardiac and pupil responses (*Allen et al., 2016*) raise the question as to whether the observed noradrenaline effects are purely a consequence of central changes, or whether peripheral effects of this drug influence metacognitive performance independently. A drug that exclusively targets peripheral, but not central, noradrenaline (cf. *De Martino et al., 2008*) could provide insight into the question of visceral contributions to metacognition as suggested in ideas on embodied cognition (*Allen and Friston, 2016*).

In conclusion, using a double-blind, placebo-controlled drug manipulation we show that noradrenaline has a controlling influence on metacognitive ability. Metacognition is enhanced following a blockade of noradrenergic $\beta$-adrenoceptors an observation suggesting potential remedial avenues for metacognitive insight deficits seen in psychiatric patients.

# Materials and methods

## Subjects

Sixty subjects participated in this double-blind, placebo-controlled, between-subjects study. Each subject was randomly allocated to one of three drug groups, controlling for an equal gender balance in all groups. Candidate subjects with a history of neurological or psychiatric disorders, current health issues, regular medications (except contraceptives), or prior allergic reactions to drugs were excluded from the study. Subjects had normal or corrected-to-normal vision. The groups were matched for age, mood (PANAS; before and after drug administration) (*Watson et al., 1988*) and intellectual abilities (WASI abbreviated version) (*Wechsler, 1999*) (*Table 1*). Subjects were reimbursed for their participation on an hourly basis. The study was approved by the UCL research ethics committee and all subjects gave written informed consent.

## Drug manipulation and procedures

To attenuate noradrenergic function we administered 40 mg of propranolol, a non-selective $\beta$-adrenoceptor antagonist. To attenuate dopamine function we administered 400 mg of amisulpride, a selective D2/3 antagonist. These drugs were chosen because of their selective high affinity effects on either one or the other of these two neuromodulators, enabling a specific dissociation of their contribution to metacognitive ability. The dosage and timing of both propranolol and amisulpride were based on previous studies that have investigated their effects on cognition (e.g., *Ramaekers et al., 1999*; *Strange et al., 2003*; *Silver et al., 2004*; *Strange and Dolan, 2004*; *Hurlemann et al., 2005*; *Alexander et al., 2007*; *Gibbs et al., 2007*; *De Martino et al., 2008*; *Kahnt et al., 2015*; *Kahnt and Tobler, 2017*).

Prior to the task the drugs were administered at two different time points, based upon pharmacokinetic considerations (*Figure 1A*). The first drug was administered 110 min prior to the metacognition task. At that time, the dopamine group received amisulpride while the other groups received placebo. After 30 min, subjects consumed a second drug. This time, the noradrenaline group received propranolol, while the dopamine and placebo group consumed a placebo. A placebo group received placebo at both times. The task was performed 80 min after the second drug administration.

## Experimental paradigm

To measure metacognitive ability we applied an adaptive visual global motion detection paradigm, similar to the version in our previous study (*Allen et al., 2016*), implemented using Psychtoolbox-3 (www.psychtoolbox.org) for MATLAB (R2010a). On every trial, subjects viewed a brief burst of motion (250 ms), followed by a forced choice to determine if the overall motion direction was to the left or right of vertical. Subjects then rated their subjective confidence using a continuous sliding scale marked at four equal intervals by horizontal lines. To prevent response preparation, the starting point of the confidence marker was jittered up to 12% to the left or right of scale midpoint on each trial. Subjects had up to 1500 ms to make their motion choice, and 2500 ms to report their confidence.

At the start of the experiment, each subject was instructed that the goal of the task was to measure their perceptual and metacognitive sensitivity. This was operationally defined as their ability to detect motion direction and how accurately their confidence ratings reflected their actual detection performance. Subjects first completed a short training session of 140 detection-only trials to establish motion thresholds. All subjects achieved staircase stabilization before continuing to the main experiment (the motion direction threshold reached in the final staircase of training was used as the starting point for main task). Subjects were encouraged to use the entire scale to report their subjective feeling of confidence, and to carefully reflect on each trial on the decision they had just made. Confidence reports were then binned into six equally sized bins for further analysis similar to previous reports (*Fleming et al., 2010*, *2012*; *Allen et al., 2017*; *Hauser et al., 2017*).

On each trial subjects viewed a cloud of 1100 moving black dots of 0.08 degrees visual angle (DVA), presented for 250 ms within a 15.69 DVA circular array at random starting positions and advancing at a speed of 0.06 DVA per frame. Dots which moved beyond the stimulus aperture were replaced at the opposite edge to maintain constant dot density. To prevent fixation on local motion

directions, all dots had a randomized limited lifetime of maximum 93% (14 frames). Each motion stimulus was defined by a global motion direction ('orientation') to the left or right of vertical. Following experiments investigating confidence with global motion stimuli, dot mean and variance were manipulated independently of one another (*Allen et al., 2016*; *Spence et al., 2016*). To this end, all dots were 'signal' dots and the standard deviation of the mean direction was adjusted across conditions. On each trial the motion signal was thus calculated using the formula:

$$\mathrm{Dot\,direction} = (\mathrm{Left\,vs\,Right}) {}^\star \mathrm{Mean\,Orientation} + \mathrm{Gaussian\,Noise} * \mathrm{Standard\,Deviation}$$

To control task difficulty and thus ensure an unbiased estimate of metacognitive sensitivity (*Fleming and Lau, 2014*), the mean direction of motion was continuously adapted for each subject using a 2-up-1 down staircase, which converges at the limit on a 71% detection accuracy. To render the staircase opaque to subjects, and maximize confidence variability, we deployed two separate staircase conditions with a fixed motion variance equal to either 20 or 30 degrees standard deviation. Subjects completed a total of 144 trials (72 high variance, 72 low variance) divided evenly between four blocks.

## Statistical analysis and metacognition modelling

The goal of our analyses were two-fold: First, we wanted to ensure all groups expressed equivalent perceptual decision making performance, as performance differences can influence estimates of metacognitive ability (*Fleming and Lau, 2014*). Second, we wanted to test for differences in metacognition using signal detection theory (SDT), examining the metacognitive detection performance using an area under the curve for a type-II receiver-operating-characteristics (ROC) (AUROC2) metric (*Fleming et al., 2010*, *2012*).

To ensure homogeneous performance across all subjects, we excluded two outlier subjects who performed worse than the rest (as measured using boxplots; both subjects belonged to the amisulpride group). We omitted all trials from the first block to ensure staircase stabilisation, similar to previous studies (*Fleming et al., 2010*, *2012*; *Allen et al., 2017*; *Hauser et al., 2017*). Trials with early (<100 ms), late (>1500 ms) or missing responses were excluded. We collapsed low and high variance trials, leaving a total of 108 trials per subject, per recommended procedures, for optimal estimation of SDT measures (*Fleming and Lau, 2014*). To compare perceptual abilities between groups, we assessed their performance in terms of accuracy and signal strength (mean stimulus orientation). We further assessed reaction times and the subjects' average confidence ratings. We used ANOVAs with a between-subject factor group (placebo, propranolol, amisulpride) and post-hoc t-tests in SPSS (version 22, IBM). To evaluate the evidence for the null hypothesis that amisulpride had no effect on AUROC2, we performed a Bayesian two-sample t-test using version 0.9.8 of the BayesFactor package, computed using R version 3.3.2 (2016-10-31) on x86_64-w64-mingw32 (*Rouder et al., 2009*; *Morey and Rouder, 2015*). A unit-information Bayes Factor with r scale parameter = 1 was calculated for the Placebo vs Amisulpride contrast (*Rouder et al., 2009*). This Bayes factor expresses the continuous evidence for the null hypothesis of no drug effect, where Bayes Factors > 3 correspond to 'moderate' evidence (*Rouder et al., 2009*; *Dienes, 2014*).

To examine subjects' metacognitive abilities, we assessed the type-II performance using SDT. In this framework, metacognition can be modelled as the sensitivity of subjective confidence to underlying ground truth discrimination performance. By defining metacognitive 'hits' (i.e., high confidence for correct detections) and 'misses' (high confidence for incorrect detections), metacognitive sensitivity can be expressed as the area under a type-II receiver-operating-characteristics curve (AUROC2). In contrast to classical measures of metacognition (e.g., the correlation of confidence and accuracy), AUROC2 is unbiased by a subject's overall level of confidence (or metacognitive bias/criterion) if detection performance is held constant across subjects (*Fleming and Lau, 2014*). Further, being nonparametric, AUROC2 is not susceptible to issues such as non-normal confidence distributions.

AUROC2 was calculated using the same metric as in *Fleming et al. (2010)*, *Kornbrot (2006)*:

$$AUROC2 = \frac{1}{4}\sum_{k=1}^{\frac{1}{2}i}\left([h_{k+1}-f_k]^2 - [h_k - f_{k+1}]^2\right) + \frac{1}{4}\sum_{k=\frac{1}{2}i}^{i}\left([h_{k+1}-f_k]^2 - [h_k - f_{k+1}]^2\right) \tag{1.1}$$

where $i$ indicates the six confidence rating bins, $h$ depicts the relative frequency of this rating for correct choices ($h_i = p(confidence == i | correct)$) and $f$ describes the counterpart for incorrect responses ($f_i = p(confidence == i | incorrect)$).

To ensure that the groups did not differ in their type-I detection performance ($d'$) or response bias ($c$), we additionally examined these metrics (*Green and Swets, 1966*; *Fleming et al., 2010*):

$$d' = \frac{1}{\sqrt{2}}(z(H) - z(FA)) \qquad (1.2)$$

$$c = -.5(z(H) - z(FA)) \qquad (1.3)$$

where $z$ describes the inverse of a cumulative normal distribution, $H$ is the correct hits and $FA$ the false alarms for two-alternative forced choice tasks.

## Acknowledgements

RJD holds a Wellcome Trust Senior Investigator Award (098362/Z/12/Z). A Wellcome Trust Cambridge-UCL Mental Health and Neurosciences Network grant (095844/Z/11/Z) supported RJD and TUH. GR and MA were supported by a Wellcome Trust SRF grant (100227). The UCL-Max Planck Centre is a joint initiative supported by UCL and the Max Planck Society. The Wellcome Trust Centre for Neuroimaging is supported by core funding from the Wellcome Trust (091593/Z/10/Z). We thank Gita Prabhu for the support with the ethics application. We thank Robb Rutledge, Francesco Rigoli, Geert-Jan Will, and Eran Eldar for helpful inputs on the study design. We thank Steve Fleming for helpful comments on an earlier version of the manuscript. The authors declare no conflict of interest.

## Additional information

### Funding

| Funder | Grant reference number | Author |
| --- | --- | --- |
| Wellcome | 095844/Z/11/Z | Tobias U Hauser<br>Michael Moutoussis<br>Raymond J Dolan |
| Wellcome | 100227 | Micah Allen<br>Geraint Rees |
| UCLH Biomedical Research Council | | Michael Moutoussis |
| Wellcome | 091593/Z/10/Z | Geraint Rees<br>Raymond J Dolan |
| Wellcome | 098362/Z/12/Z | Raymond J Dolan |
| Max-Planck-Gesellschaft | | Raymond J Dolan |

The funders had no role in study design, data collection and interpretation, or the decision to submit the work for publication.

### Author contributions

TUH, Conceptualization, Data curation, Formal analysis, Supervision, Visualization, Writing—original draft, Project administration, Writing—review and editing; MA, Conceptualization, Software, Supervision, Methodology, Writing—original draft, Writing—review and editing; NP, Investigation, Project administration, Writing—review and editing; MM, Conceptualization, Supervision, Investigation, Methodology, Writing—review and editing; GR, Conceptualization, Supervision, Writing—review and editing; RJD, Conceptualization, Supervision, Funding acquisition, Writing—original draft, Writing—review and editing

### Author ORCIDs

Tobias U Hauser, http://orcid.org/0000-0002-7997-8137
Micah Allen, http://orcid.org/0000-0001-9399-4179

Michael Moutoussis, http://orcid.org/0000-0002-4751-0425
Geraint Rees, http://orcid.org/0000-0002-9623-7007
Raymond J Dolan, http://orcid.org/0000-0001-9356-761X

**Ethics**
Human subjects: The study was approved by the UCL research ethics committee and all subjects gave written informed consent.

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
