## [Decision Letter]

Thank you for submitting your article "Noradrenaline, but not dopamine, modulates metacognitive ability". Your article has been reviewed by two peer reviewers, one of whom is a member of our Board of Reviewing Editors, and the evaluation process was overseen by Sabine Kastner as Senior Editor. The reviewers were supportive of the manuscript, but recommended the following major revisions, before a decision on publication can be made.

In the current submission, the authors put forth a clearly designed behavioral task to probe the role of noradrenaline and dopamine on perceptual confidence. Using amisulpride and propranolol as dopamine/noradrenaline blockers, the authors reported that administration of propranolol affected perceptual confidence judgment in the subjects, while that of amisulpride did not. The reviewers' biggest concern is one that is common with studies using pharmacological manipulations: the two agents used have effects other than the ones desired in the study, and they have different pharmacokinetics and dosage effects. The authors gave some considerations to these factors, but there are still uncontrolled aspects. The timeframe of the administration, the dosage of the two drugs, and the other targets of the drugs are factors that concern the reviewers the most. The authors are advised to discuss these factors further and provide justification for the practice in the study, or tone down the thesis to a comparison between propranolol and amisulpride administration, not noradrenaline and dopamine.

Reviewer #1:

In this Research Advance article, Hauser and colleagues presented an elegant and straightforward experiment, probing roles of dopamine noradrenaline in metacognition. Administration of noradrenaline but not dopamine affects metacognitive judgment accuracy but not the cognitive performance itself. The results pave a road for further investigations into how visceral senses influence metacognition. The manuscript is overall clearly written and informative. However, there are a few issues to be addressed.

First, the timing of drug administration is different in the noradrenaline group and the dopamine group. The authors stated that the motivation is that amisulpride and propranolol have different kinetics, which is reasonable. However, is there any control method to make sure that these two different schedules are equivalent to each other? In other words, if propranolol was injected further ahead of the testing, could it have had an effect? The authors should address the potential discrepancies here. (While the authors do cite previous studies as motivation for these specific times in the methods section, note that its metacognition, not cognition, that's investigated here; the authors themselves have argued in the Introduction that these two may have different mechanisms. Do findings regarding the kinetics of these drugs still apply in a metacognition study?)

Second, pharmacological agents are never precise effectors (e.g. the high affinity issue). Instead they may have side effects that cause issues. What are some of the effects amisulpride and propranolol may have, how strong they are at the doses used, and how may they confound interpretation of the results?

Finally, could any of the findings regarding metacognition be task-specific? For instance, could non-perceptual, more slow-paced decision making have different interactions with the drugs? The authors should include a discussion.

Reviewer #2:

This paper reports the effects of single dose administration of propranolol or amisulpride on self-rated confidence following perceptual decision making. In a double blind between-subjects design, they found that administration of propranolol increased the likelihood that participants assigned low confidence to decisions that were wrong. Amisulpride had no effect on self-rated confidence. Neither drug influenced perceptual performance. These results are framed in the context of the influence of noradrenaline and dopamine on meta-cognition.

The behavioural study is well designed and the results clearly reported. My concerns relate mostly to the pharmacological inferences and the framing of the paper.

1) Amilsulpride did not significantly influence performance (perceptual or self-rated confidence). It is quite possible that this null effect is simply due to the low occupancy of dopamine receptors with this single dose. To support this dose, the authors' cited paper (Kahnt et al., 2015) in turn cites Mehta (2008) whom in turn use PET (raclopride) to show that a single dose of 400mg of amilsupride leads to an estimated 28% receptor occupancy (much lower than clinical effects) which notably caused no significant changes across a broad armoury of cognitive tests (had they properly corrected for multiple tests!). This is consistent with Ramaekers (1999) – cited by Gibbs 2007 to justify the dose – who found no cognitive or behavioural effects following a single dose unless it was administered for 5 days in a row.

We do not know what the comparable receptor occupancy of central adrenoreceptors is following a single dose of 40mg of propranolol (there are no citations in the text and I believe there is no PET assay). Therefore, the apparent head-to head competition between noradrenaline and dopamine modulation is not unequivocally demonstrated. The difference could simply be due to a marked difference in the central effective pharmacological effects of these single doses. This might be fine if the paper framed itself around the actual pharmacological manipulations given – but the paper clearly stamps itself as a dopamine versus noradrenaline head-to-head. The authors offer insufficient justification for this imputation and list no caveats or limitations.

2) As I'm sure the authors are acutely aware, propranolol has both central and peripheral β blockade – in fact prior work by some of the authors used propranolol in combination with the selective peripheral blocker, nadolol, to disambiguate the central from the peripheral action. So there are inevitably cardiovascular and likely other autonomic consequences to the β blockade. Given the same group recently published a paper that linked autonomic fluctuations to decision confidence, this seems quite an oversight.

3) There is more of a style issue, but I found the start of the Discussion a much more accessible/more descriptive account of the goals of the paper, than the title or the Abstract – that is, I would have preferred to see a more descriptive approach to the actual task up front – self-rated decision following a perceptual task which was then interpreted as "metacognition". The latter is a conceptual construct that must surely be composed of other processes, and which is only (imperfectly) addressed in the present task. The broad readership will have to read deeply into the paper until they discover that this is a simple rating of confidence following a decision.

4) Were the subjects fasting? Were there checks to see if they were able to unblind themselves (particularly for the propranolol).

5) Was there a main effect of drug on PANAS?

Citations:

Mehta, M. A., Montgomery, A. J., Kitamura, Y., & Grasby, P. M. (2008). Dopamine D2 receptor occupancy levels of acute sulpiride challenges that produce working memory and learning impairments in healthy volunteers. Psychopharmacology, 196(1), 157-165.

Ramaekers JG, Louwerens JW, Muntjewerff ND, Milius H, de Bie A, Rosenzweig P, Patat A, O'Hanlon JF: Psychomotor, cognitive, extrapyramidal, and affective functions of healthy volunteers during treatment with an atypical (amisulpride) and a classic (haloperidol) antipsychotic. J Clin Psychopharmacology 1999; 19:209-221

---

## [Author Response]

*Reviewer #1:*

*In this Research Advance article, Hauser and colleagues presented an elegant and straightforward experiment, probing roles of dopamine noradrenaline in metacognition. Administration of noradrenaline but not dopamine affects metacognitive judgment accuracy but not the cognitive performance itself. The results pave a road for further investigations into how visceral senses influence metacognition. The manuscript is overall clearly written and informative. However, there are a few issues to be addressed.*

*First, the timing of drug administration is different in the noradrenaline group and the dopamine group. The authors stated that the motivation is that amisulpride and propranolol have different kinetics, which is reasonable. However, is there any control method to make sure that these two different schedules are equivalent to each other? In other words, if propranolol was injected further ahead of the testing, could it have had an effect? The authors should address the potential discrepancies here. (While the authors do cite previous studies as motivation for these specific times in the methods section, note that its metacognition, not cognition, that's investigated here; the authors themselves have argued in the Introduction that these two may have different mechanisms. Do findings regarding the kinetics of these drugs still apply in a metacognition study?)*

We thank the reviewer for a positive evaluation of the manuscript and for raising thoughtful questions. We agree that directly comparing two drugs comes with specific challenges, some of which cannot be completely resolved with current methods. In this study, we took great care to make the drug conditions as comparable as possible. First, we tried to match the pharmacokinetics between the drugs. For propranolol, the peak plasma concentration is reached 1-2 hours after drug administration and most experimental studies are conducted approx. 90 minutes after drug administration (usually between 75 and 120 minutes) (cf. Alexander et al., 2007; De Martino et al., 2008; Hurlemann et al., 2005; Silver et al., 2004; Strange et al., 2003; Strange and Dolan, 2004). For amisulpride, the peak plasma concentration is between 1-4 hours, and experimental tasks are usually delivered any time between 1 and 3 (up to 6) hours after administration (cf. Gibbs et al., 2007; Kahnt et al., 2015; Kahnt and Tobler, 2017; Peretti et al., 1997; Ramaekers et al., 1999). Based on these established facts and procedures, we opted to administer amisulpride ahead of administering propranolol. However, given the relatively long duration of peak plasma levels (i.e. relatively slow metabolisation), we think it is reasonable to assume we would obtain similar results with slightly different administration regiments.

We used similar reasoning for selecting dosages for the two drugs. Based on prior literature, 40mg of propranolol as well as 400mg of amisulpride can be considered standard dosages in the context of experimental studies and where these dosages are known to impact cognition and neural processing (e.g., Alexander et al., 2007; De Martino et al., 2008; Kahnt et al., 2015; Kahnt and Tobler, 2017; Ramaekers et al., 1999; Strange et al., 2003; Strange and Dolan, 2004). However, little is known about whether these dosages are comparable in terms of their effect on the brain. To our knowledge, there are no PET studies on single-dose drug administration that could allow us to quantify the extent to which these drug dosages occupy their respective receptors. And even if we had access to such knowledge it is unclear whether (and how) receptor occupancy linearly translates to effects on cognition. This means we cannot say with certainty whether an absence of an amisulpride effect is because dopamine does not play a role in metacognition, or whether it is because the dosage was too little. We have thus revised the manuscript to acknowledge this possibility, including toning down our findings with respect to amisulpride, and mention these issues explicitly in the revised discussion. Please find the revised sections below.

Lastly, it is worth mentioning that we would not expect different pharmacokinetics for cognition and metacognition. Based on the assumption that both processes are derive from brain-based computations, then there is good reason to believe that pharmacokinetics of both metacognition and cognition should be equivalent.

Title: “Noradrenaline blockade specifically enhances metacognitive performance”

Abstract: “Blockade of dopamine D2/3 receptors (400 mg amisulpride) had no effect on either metacognition or perceptual decision making.”

Introduction: “[…] we selected drugs with selective high affinity, the β-adrenoceptor antagonist propranolol in the case of noradrenaline, and the D2/3 receptor antagonist amisulpride in the case of dopamine. In a double-blind, placebo-controlled design we demonstrate that the noradrenergic agent propranolol uniquely improves metacognition in the absence of an effect on perceptual performance, with no effect seen following administration of the dopamine antagonist amisulpride.”

Results: “Noradrenaline blockade modulates metacognition […] A dopamine subject group first received 400mg amisulpride (dopamine D2/3 receptor antagonist) and subsequently placebo, whereas the noradrenaline group first received placebo and then 40mg propranolol (β-adrenoceptor antagonist). […] These results indicate that antagonism of noradrenergic function improves metacognitive insight, in the absence of any effect of amisulpride. […] A highly significant effect of propranolol compared to placebo shows that propranolol increases metacognitive abilities. The difference between propranolol and amisulpride suggests that this performance increase might be specific to an influence on noradrenaline but not dopamine function.”

Discussion: “Here we show that inhibition of central noradrenaline (by means of propranolol) function enhances perceptual metacognitive ability. A dopamine blockade (by means of amisulpride) had no impact on metacognition and neither drug manipulation had an impact on core perceptual performance. […] In our experiment, perceptual metacognition was solely influenced by manipulation of noradrenaline, and not by blocking dopamine D2/D3 receptors. […] An important caveat for comparing the amisulpride and propranolol groups directly is that little is known about the precise pharmacokinetics and how comparable the dosage effects are. We took great care in the design of the study to render the two drug conditions as comparable as possible. First, because of the slightly different absorption rates, we administered amisulpride 30 minutes before propranolol, in keeping with previous drug schedules (Peretti et al., 1997; Ramaekers et al., 1999; Strange et al., 2003; Silver et al., 2004; Strange and Dolan, 2004; Hurlemann et al., 2005; Alexander et al., 2007; Gibbs et al., 2007; De Martino et al., 2008; Kahnt et al., 2015; Kahnt and Tobler, 2017). Second, to render the cognitive effects of the drugs as similar as possible, we selected dosages that were commonly reported in previous studies of neurocognition (i.e. 40 mg propranolol, 400mg amisulpride) (e.g., Ramaekers et al., 1999; Strange et al., 2003; Silver et al., 2004; Strange and Dolan, 2004; Hurlemann et al., 2005; Alexander et al., 2007; Gibbs et al., 2007; De Martino et al., 2008; Kahnt et al., 2015; Kahnt and Tobler, 2017). However, we know little about the magnitude of these drug effects on the brain. A previous study of sulpiride, which has a similar chemical formulation to amisulpride, but slightly different pharmacokinetics, suggests that a single-dose of 400mg leads to a occupancy of ~28% of D2 receptors (Mehta et al., 2008). Unfortunately, there are no PET studies reporting on single-dose amisulpride, and there are no occupancy studies of propranolol, thus rendering it difficult to directly quantify and compare our dosage effects. […] In conclusion, using a double-blind, placebo-controlled drug manipulation we show that noradrenaline has a controlling influence on metacognitive ability.”

*Second, pharmacological agents are never precise effectors (e.g. the high affinity issue). Instead they may have side effects that cause issues. What are some of the effects amisulpride and propranolol may have, how strong they are at the doses used, and how may they confound interpretation of the results?*

Common side effects of amisulpride are extrapyramidal symptoms and dizziness/nausea, but these are rarely observed in single-dose administration regimes such as ours. In fact, a previous study did not find any significant effects on psychomotor or extrapyramidal functioning after 400mg amisulpride administration (Ramaekers et al., 1999). In this study, we had one participant in the amisulpride group that complained of mild transient symptoms (primarily tremor) in the aftermath of the experiment, likely to be due to the drug. For propranolol, common side effects are lowered blood pressure and pulse, nausea, and diarrhoea. This is why we examined blood pressure and pulse after the experiment. We did not observe any differences in blood pressure (systolic: F(2,57)=.53, p=.594, diastolic: F(2,57)=.825, p=.443), but observed a trend difference in pulse (F(2,57)=2.95, p=.060). However, the pulse rates in all groups were still within the normal range (placebo: 62.9 ± 8.6, propranolol: 56.3 ± 11.1, amisulpride: 62.1 ± 8.3). No subject reported issues related to low blood pressure at the end of the experiment. In addition, the effect on metacognition was much stronger than the effect on pulse, and the metacognitive effect remained even when controlling for pulse (multiple regression placebo vs propranolol: t=4.28, p<.001). Nevertheless, we think it would be interesting to investigate whether peripheral noradrenaline effects also influence metacognitive performance, and discuss possible ways of addressing this in future studies.

Discussion: “Lastly, our previous findings of an embodied reflection of confidence by means of cardiac and pupil responses (Allen et al., 2016) raises the question as to whether the observed noradrenaline effects are purely a consequence of central changes, or whether peripheral effects of this drug influence metacognitive performance independently. A drug that exclusively targets peripheral, but not central, noradrenaline (cf. De Martino et al., 2008) could provide insight into the question of visceral contributions to metacognition as suggested in ideas on embodied cognition (Allen and Friston, 2016).”

*Finally, could any of the findings regarding metacognition be task-specific? For instance, could non-perceptual, more slow-paced decision making have different interactions with the drugs? The authors should include a discussion.*

This is an interesting issue, which is currently under discussion in the field. Our reading of the literature is that metacognitive performance is relatively stable across different tasks within the domain of perceptual decision making (e.g., Garfinkel et al., 2016; McCurdy et al., 2013; Song et al., 2011), even between different sensory modalities (Gardelle et al., 2016). However, there also seem to be some differences between cognitive domains, e.g., between perceptual decision making and meta-memory (Baird et al., 2013; Fleming et al., 2014). Given that we have only evaluated perceptual but not other domains of metacognition, we cannot say whether the noradrenaline effect would generalise. We have thus clarified throughout the revised manuscript that we study perceptual decision making, and discuss this issue in the Discussion section.

Discussion: “Here we show that inhibition of central noradrenaline (by means of propranolol) function enhances perceptual metacognitive ability […] In this study, we show that noradrenaline specifically influences perceptual metacognition but not perceptual decision making. It is interesting to speculate whether our findings are generalizable to metacognition in non-perceptual domains. Recent studies show that a metacognitive ability is relatively stable across different perceptual decision making tasks (Song et al., 2011; McCurdy et al., 2013), even when probing different sensory modalities (Gardelle et al., 2016; Garfinkel et al., 2016). However, it is unclear whether metacognition within different cognitive domains (e.g., perception vs memory) rely on the same processes, with evidence from neuroimaging suggesting that these functions utilise unique neural networks (Baird et al., 2013; Fleming et al., 2014). Given that noradrenaline modulates activity on a whole-brain level (Hauser et al., 2016), it is possible that noradrenergic metacognition effects can be observed in domains other than perception, an interesting area for future studies.”

*Reviewer #2:*

[…]

*1) Amilsulpride did not significantly influence performance (perceptual or self-rated confidence). It is quite possible that this null effect is simply due to the low occupancy of dopamine receptors with this single dose. To support this dose, the authors' cited paper (Kahnt et al., 2015) in turn cites Mehta (2008) whom in turn use PET (raclopride) to show that a single dose of 400mg of amilsupride leads to an estimated 28% receptor occupancy (much lower than clinical effects) which notably caused no significant changes across a broad armoury of cognitive tests (had they properly corrected for multiple tests!). This is consistent with Ramaekers (1999) – cited by Gibbs 2007 to justify the dose – who found no cognitive or behavioural effects following a single dose unless it was administered for 5 days in a row.*

*We do not know what the comparable receptor occupancy of central adrenoreceptors is following a single dose of 40mg of propranolol (there are no citations in the text and I believe there is no PET assay). Therefore, the apparent head-to head competition between noradrenaline and dopamine modulation is not unequivocally demonstrated. The difference could simply be due to a marked difference in the central effective pharmacological effects of these single doses. This might be fine if the paper framed itself around the actual pharmacological manipulations given – but the paper clearly stamps itself as a dopamine versus noradrenaline head-to-head. The authors offer insufficient justification for this imputation and list no caveats or limitations.*

We thank the reviewer for the positive evaluation of our work and we appreciate the helpful critique related to what inferences we can draw based upon our drug manipulations. First, we 400mg of amisulpride has been used by several prior studies and we decided on this dosage because it is a fairly standard dose for non-clinical, experimental studies (e.g., Gibbs et al., 2007; Kahnt et al., 2015; Kahnt and Tobler, 2017; Ramaekers et al., 1999). Several studies have shown effects of this dose on cognitive and neural responses. However, we agree that there are also several null-results with respect to cognition in the literature, but it is difficult to reconcile whether this is due to a low dosage or because D2 receptors do not play a major role in the particular aspect of cognition under investigation. In addition, our goal was not to relate it to clinical doses as used in the treatment of conditions such as schizophrenia. We had a more fundamental question of whether amisulpride influences metacognition. The study by Mehta et al., 2008 used sulpiride instead of amisulpride. Although these two drugs are closely related, they do have different pharmacokinetic properties, so it is not straightforward to generalise from this study as the issue of the degree of receptor occupancy following a 400mg amisulpride dose. To the best of our knowledge, there is no study that investigates receptor occupancy of 400mg of amisulpride in healthy controls (single-dose), rendering it difficult to guage the extent of dopamine receptor occupancy with this dose. However, given previously mentioned effects on cognition and neural responses, an effect on perceptual decision making or metacognition is entirely possible. Nevertheless, we agree with the reviewer that a direct comparison between the drugs is difficult to interpret because of multiple factors that might confound such a comparison (for example, dosage). We revised the manuscript accordingly and toned down the dopamine aspects (e.g., removed it from the title). In addition, we cite the aforementioned papers and discuss issues of interpretation in a new Discussion section.

*2) As I'm sure the authors are acutely aware, propranolol has both central and peripheral β blockade – in fact prior work by some of the authors used propranolol in combination with the selective peripheral blocker, nadolol, to disambiguate the central from the peripheral action. So there are inevitably cardiovascular and likely other autonomic consequences to the β blockade. Given the same group recently published a paper that linked autonomic fluctuations to decision confidence, this seems quite an oversight.*

We agree that a dissociation between central and peripheral effects of noradrenaline is both of interest and important. However, to have a comparable control condition for both, amisulpride and propranolol, we decided to use placebo instead of a peripherally active β-blocker. Our reading of the literature on propranolol studies led us to conclude placebo is much more common as a control condition than is nadolol (where a dosage issue again arises). Note also that our previous finding (Allen et al., 2016) showed that confidence is reflected in peripheral measures. However, it is still entirely unclear whether this is a down-stream effect of a brain response (very likely for pupil dilation), or whether it is an independent peripheral effect. Importantly, as there is no prior study the effects of noradrenaline on metacognition, our prime goal was to determine whether there were any effects of noradrenaline before examining differential responses to central and peripheral noradrenaline blockade. In subsequent studies, we indeed plan to use this differentiation to investigate the specific contributions of embodied cognition. We discuss this issue now in the revised version of the manuscript.

Discussion: “[…] our previous findings of an embodied reflection of confidence by means of cardiac and pupil responses (Allen et al., 2016) raises the question as to whether the observed noradrenaline effects are purely a consequence of central changes, or whether peripheral effects of this drug influence metacognitive performance independently. A drug that exclusively targets peripheral, but not central, noradrenaline (cf. De Martino et al., 2008) could provide insight into the question of visceral contributions to metacognition as suggested in ideas on embodied cognition (Allen and Friston, 2016)”

*3) There is more of a style issue, but I found the start of the Discussion a much more accessible/more descriptive account of the goals of the paper, than the title or the Abstract – that is, I would have preferred to see a more descriptive approach to the actual task up front – self-rated decision following a perceptual task which was then interpreted as "metacognition". The latter is a conceptual construct that must surely be composed of other processes, and which is only (imperfectly) addressed in the present task. The broad readership will have to read deeply into the paper until they discover that this is a simple rating of confidence following a decision.*

We apologise for not being clear enough about the concept of metacognition and the analysis used in this study. We have revised the manuscript entirely to make the beginning of the paper more accessible for the broad readership of *eLife* and to clarify the meaning of metacognition right at the beginning of the paper. We hope that the reviewer agrees that the revised version is now more accessible.

Abstract: “Impairments in metacognition, the ability to accurately report one’s performance, are common in patients with psychiatric disorders, where a putative neuromodulatory dysregulation provides the rationale for pharmacological interventions.”

Introduction: “Making a decision is often accompanied by a conscious feeling of confidence (Flavell, 1979). Subjective confidence reports typically show a good correspondence to actual task performance, reflecting a metacognitive ability for accurate introspection (Fleming et al., 2010). Impairments in metacognition can compromise decision making and lead to misjudgements of actual performance, as found in several psychiatric dimensions […].”

*4) Were the subjects fasting? Were there checks to see if they were able to unblind themselves (particularly for the propranolol).*

We did not ask the subjects to fast, as in similar drug studies (e.g., Kahnt et al., 2015, 2017; Strange & Dolan, 2004), but they did not eat anything during the period of the entire study. We asked the subjects after the experiment to guess which drug they received (missing data from two subjects). Based on the reviewer’s comments, we now analysed this data and did not find any difference. We report this in the revised version of the manuscript and detail here the subjects’ guesses.

drug guessed (N)placebopropranololamisulpridedrugs received (N)placebo864propranolol965amisulpride893

Results: “A post-experiment evaluation revealed that subjects were not aware of whether and which drug they received (χ2(4)=1.26, p=.868; missing data from two subjects).”

*5) Was there a main effect of drug on PANAS?*

No, as reported in Table 1 there was no effect on positive or negative affect. We extended these analyses in the revised manuscript by reporting a repeated-measures ANOVA with within-subjects affect (positive, negative), time (pre, post), and between-subjects factor drug. This extended analysis also showed no effects of drug on mood.

Results: “There were no effects of drug on mood (PANAS; Watson et al., 1988) ratings (main effect of drug: F(2,57)=.16, p=.852; time x drug: F(2,57)=.19, p=.827; time x affect x drug: F(2,57)=2.17, p=.124).”